# Co-Treatment of Copper Oxide Nanoparticle and Carbofuran Enhances Cardiotoxicity in Zebrafish Embryos

**DOI:** 10.3390/ijms22158259

**Published:** 2021-07-31

**Authors:** Ferry Saputra, Boontida Uapipatanakul, Jiann-Shing Lee, Shih-Min Hung, Jong-Chin Huang, Yun-Chieh Pang, John Emmanuel R. Muñoz, Allan Patrick G. Macabeo, Kelvin H.-C. Chen, Chung-Der Hsiao

**Affiliations:** 1Department of Bioscience Technology, Chung Yuan Christian University, Chung-Li 320314, Taiwan; g10865013@cycu.edu.tw; 2Department of Chemistry, Faculty of Science and Technology, Rajamangala University of Technology, Thanyaburi 12110, Thailand; boontida_u@rmutt.ac.th; 3Department of Applied Physics, National Pingtung University, Pingtung 900391, Taiwan; jslee@mail.nptu.edu.tw (J.-S.L.); a0978038385@gmail.com (S.-M.H.); 4Department of Applied Chemistry, National Pingtung University, Pingtung 900391, Taiwan; hjc@mail.nptu.edu.tw (J.-C.H.); jojo89120422@gmail.com (Y.-C.P.); 5Laboratory for Organic Reactivity, Discovery and Synthesis (LORDS), Research Center for the Natural and Applied Sciences, University of Santo Tomas, Manila 1008, Philippines; johnemmanuel.munoz.sci@ust.edu.ph; 6Department of Chemistry, Chung Yuan Christian University, Chung-Li 320314, Taiwan; 7Center for Nanotechnology, Chung Yuan Christian University, Chung-Li 320314, Taiwan; 8Research Center for Aquatic Toxicology and Pharmacology, Chung Yuan Christian University, Chung-Li 320314, Taiwan

**Keywords:** CuO nanoparticle, carbofuran, zebrafish, cardiotoxicity, molecular docking

## Abstract

The use of chemicals to boost food production increases as human consumption also increases. The insectidal, nematicidal and acaricidal chemical carbofuran (CAF), is among the highly toxic carbamate pesticide used today. Alongside, copper oxide nanoparticles (CuO) are also used as pesticides due to their broad-spectrum antimicrobial activity. The overuse of these pesticides may lead to leaching into the aquatic environments and could potentially cause adverse effects to aquatic animals. The aim of this study is to assess the effects of carbofuran and copper oxide nanoparticles into the cardiovascular system of zebrafish and unveil the mechanism behind them. We found that a combination of copper oxide nanoparticle and carbofuran increases cardiac edema in zebrafish larvae and disturbs cardiac rhythm of zebrafish. Furthermore, molecular docking data show that carbofuran inhibits acetylcholinesterase (AChE) activity in silico, thus leading to impair cardiac rhythms. Overall, our data suggest that copper oxide nanoparticle and carbofuran combinations work synergistically to enhance toxicity on the cardiovascular performance of zebrafish larvae.

## 1. Introduction

The increased need for pesticides in agricultural industries has become a global concern to the aquatic environment. While the use of pesticides enhances crop productivity, it is considered harmful to the environment especially to aquatic environments where wastes can accumulate. Pesticides can enter the aquatic system by direct application, spray drift, aerial spraying, precipitation from the atmosphere, or through run-offs from agricultural areas [1]. They can also enter the food chain and bioaccumulate in aquatic animals and further comes back to humans as food [2,3,4,5]. In recent years, studies have proven that pesticides can have adverse effects on aquatic animals. Several studies suggest that pesticide toxicity might be caused by the change in the balance between the production of reactive oxygen species (ROS) and antioxidant activity [6,7,8].

Carbofuran is among the highly toxic carbamate pesticides mainly used as insecticide, nematicide, and acaricide [9]. Despite being strictly banned for many decades and strictly regulated in several countries, the carbofuran in the agricultural sector remains practiced. In 2015, carbofuran is the third highest pesticide residue found in fruits and vegetables in Taiwan [10]. Carbofuran works by inhibiting acetylcholinesterase activity in the brain causing alterations in behavior [11,12,13]. Carbofuran also induces oxidative stress in the brain of mammals by promoting lipid peroxidation enabling free radical production [14,15]. Relevant to this study, carbofuran increases oxidative stress in zebrafish [16,17]. The brain regulates physiological functions thus a damage caused by ROS in the brain, which has low mitotic rate and relatively low antioxidant activity, is fatal to humans [15,18].

In addition to carbamate pesticides, heavy metal nanoparticles have also emerged as new pollutants to the aquatic environment. Heavy metals are persistent in the environment and are highly toxic to aquatic environments. Heavy metals become toxic when they cannot be metabolized in the body and are accumulated in soft tissues [19,20]. Copper nanoparticles for example are one of the heavy metal nanoparticles that are commonly used in agricultural industries alongside gold, silver, zinc, aluminum, and silica nanoparticle [21]. In aquicultural industries, copper nanoparticles are commonly used due to their availability and broad-spectrum antimicrobial activity [22]. Although copper is a trace element that is needed for body function, too much copper causes copper poisoning. Moreover, copper in tissues of marine animals has been established as an environmental pollutant and harmful to humans as well as aquatic organisms [23,24,25,26]. Many studies also suggest that copper induces oxidative stress and cardiotoxicity making copper toxicity a global concern [27,28,29]. Thus, assessment of aquatic toxicity propels the use of a sensitive animal model.

In this study, we propose the use of *Danio rerio* (zebrafish) as animal model. The use of zebrafish as animal model for determining aquatic toxicity has been demonstrated due to its excellent sensitivity to change of water quality [30]. Furthermore, the zebrafish genome has been sequenced and more than 70% of zebrafish genes have been reported to be homologous with human genes [31]. The majority of human dilated cardiac pathways have corresponding homologs in zebrafish genes [32]. In addition, zebrafish signaling pathways and functions are also similar with mammals compared to other aquatic animal models [33].

Under usual environmental conditions, the pollutant may consist of different kinds of substances and may exhibit ecotoxic effects originate especially from mixture of chemicals found in estuaries where pollutants are pooled together [34,35]. A previous study demonstrated that copper nanoparticles may interact with pesticides to increase persistence of herbicides in the environment [36]. In this study, the synergistic effects of copper and carbofuran in aquatic zebrafish larvae after incubation was investigated through assessment of cardiac physiology and rhythm. Thus, insights as to the damage caused by multiple pollutants to aquatic animals are disclosed in this study.

## 2. Results

### 2.1. Analysis of Physical Properties of Copper Oxide Nanoparticles

The X-ray diffraction (XRD) pattern of the test sample was in good agreement with monoclinic CuO phase with a tenorite structure (JCPDS 89-2529) (Figure 1A). No additional crystalline phase apart from the CuO was detected. The X-ray pattern contained two sharp peaks at 2θ = 35.6 and 38.8°, which matched reflections from (-111)/(002) and (111) planes of tenorite, respectively. Average crystallite size (D_111_) was estimated to be 14.7 nm using the Scherrer equation [37]. According to the Raman analysis, three Raman peaks around 293, 333 and 616 cm^−1^ were noted in the test sample (Figure 1B). These Raman peaks corroborates to CuO data described in the literature (288, 330 and 621 cm^−1^) [38]. With reference to the previously reported powder and single crystal Raman spectra of CuO [39,40,41], the peak at 293 cm^−1^ was readily assigned to Ag and the peaks at 333 and 616 cm^−1^ to the Bg modes, respectively. The Fourier-transform infrared (FTIR) pattern of the test sample showed major absorption bands at 3418, 1647, 1590, 1399, and 550 cm^−1^ (Figure 1C). The absorption around 3418 cm^−1^ may be attributed to the stretching vibration of absorbed water or surface hydroxyls of the sample. The bands around 1647 and 1590 cm^−1^ were due to asymmetric stretching vibrations while the absorption peak 1399 cm^−1^ is attributed to symmetrical stretching vibration of RCOO absorbed onto the particle surface, respectively [42,43]. The absorption at 550 cm^−1^ is a characteristic of the stretching vibration of Cu-O bond, in which band splitting is illustrated with increasing particle size of monoclinic CuO [42,44]. In the UV-Vis spectra obtained for monoclinic CuO particles, a broad absorption band was detected at around ~350 nm and the absorption edge gradually blue-shifted with the size decreasing [43]. As shown in Figure 1D, UV-Vis absorption spectroscopy revealed that the sample has a wide absorption peak around 300 nm indicating bigger average particle size in the aqueous CuO suspension.

TEM images revealed that CuO particles are typically flaky and elongated in shape. As shown in Figure 1E, the mean particle size *(*D_TEM_) of the test sample is 40.3 nm with a broader particle distribution. The obvious agglomeration of CuO nanoparticles is due to the fact that D_111_ is remarkably smaller than D_TEM_ in the test sample. Intensity-based particle size distribution and the mean particle size obtained by DLS analysis are also shown in Figure 1F. The results show that the D_h_ values obtained was 152.5 nm, which is bigger than the mean particle size obtained from TEM measurement, indicating the CuO sample was clearly agglomerated and thus significantly larger in size. Our measurement shows that the zeta potential for the test sample is −72.7 mV, suggesting a higher stability of the dispersion (Figure 1G). Generally, particles are considered stable with zeta potential lower than −30 mV and higher than +30 mV due to significant charge repulsions between particle dispersion in solvents [45].

### 2.2. Copper Oxide Nanoparticle and Carbofuran Induce Cardiotoxicity in Zebrafish Larvae

According to National Cancer Institute, cardiotoxicity is a toxicity that can affect the heart through changes observed in cardiac physiology [46]. In order to investigate the cardiotoxicity caused by copper oxide nanoparticle and carbofuran, cardiac physiology parameters such as stroke volume, cardiac output, heart rate, shortening fraction and ejection fraction were analyzed. In addition, the cardiac rhythm parameter was also checked as cardiac rhythm is an important factor in cardiac physiologic analysis [47]. To select the concentration used for co-treatment, several concentrations of both copper oxide nanoparticle and carbofuran were analyzed and the test concentration that exhibited significant alterations in cardiac physiology was selected for co-treatment (Figure 2).

Based on the LC50 data, sublethal concentration of 0.1, 0.3, and 0.6 mM of copper oxide nanoparticles were selected and the cardiac physiology of each group was analyzed. Compared to the control group, the group treated with copper oxide nanoparticles for 48 h exhibited a decrement in stroke volume (*p* = 0.0414) (Figure 3A) and shortening fraction (*p* = 0.0489) (Figure 3F) in a dose dependent manner with 0.6 mM concentration which showed the statistical difference. Furthermore, heart rate also decreases in dose-dependent manner with concentration of 0.1 mM enough to significantly decrease the heart rate (*p* = 0.0007 for atrium and *p* = 0.0006 for ventricle) (Figure 3B,E). These data suggest that copper oxide nanoparticles may impair the ability of the heart to contract. Some alteration in cardiac rhythm was also observed in the atrium chamber as SD1 and SD2 value of the atrium chamber was also noted to increase, which signifies that the atrium beat was getting irregular (*p* = 0.0394 for SD1) and *p* = 0.0370 for SD2) (Figure 4A,D). Significant change was also observed in the ventricle to atrium (V-A) beat interval (Figure 4F) supporting that acute exposure to copper oxide nanoparticles indeed dysregulated the cardiac rhythm of zebrafish larvae.

Compared to the copper oxide nanoparticles treated group, the carbofuran treated group showed an increment following a dose-dependent manner in several cardiac parameters like stroke volume, cardiac output, ejection fraction, and shortening fraction with a significant difference observed at a concentration of 6.78 µM for stroke volume (*p* = 0.0232) and 9.04 µM for ejection fraction (*p* = 0.0461) (Figure 5A,C,D,F). Furthermore, some arrhythmia was also noted in several fishes after exposure to 9.04 µM of carbofuran which was illustrated by the different number of heartbeats in atrium and ventricle (Figure 5B,E). The alteration in cardiac rhythm could also have been observed after carbofuran exposure as the SD1 and SD2 values of the atrium chamber increased in a dose-dependent manner and the SD1 and SD2 values of the ventricle chamber significantly reduced (*p* < 0.05) (Figure 6A,B,D,E). Although several changes in the heart rhythm were noted, no significant change was observed in the time interval between atrium to ventricle (A-V) relaxation and vice-versa up to 6.78 µM exposure (Figure 6C,F). No data were obtained at a concentration of 9.04 µM due to several fish exhibiting severe arrhythmia. Thus, acquiring A-V and V-A interval data was not possible.

Other phenotypic alterations after carbofuran exposure also could have been noted after 48-h exposure such as body malformation (data not shown) and several fish that may underwent cardiac edema, wherein occurrence rate was high especially at higher concentrations of carbofuran (>4.25 µM) (Figure A3).

Based on the single exposure data of copper nanoparticle and carbofuran on cardiac physiology, we noted that both have contradicting effects in some cardiac physiology parameters such as stroke volume, cardiac output, shortening fraction, and ejection fraction. However, both have the same effect in dysregulating cardiac rhythm. Thus, we investigated whether both compounds will synergistically work together in causing cardiotoxicity.

Selection of the copper oxide nanoparticle and carbofuran concentration was based on the single exposure data. After incubation in a combination of 0.6 mM of copper oxide nanoparticles and 9.04 µM of carbofuran, no significant decrement was observed in the heart rate of zebrafish after co-incubation with both test substances, showing that the effect of copper nanoparticles in reducing the heart rate was lowered after co-incubation in both compounds (*p* > 0.05) (Figure 7B,E). Furthermore, other parameters such as stroke volume, cardiac output, and shortening fraction were also not significantly different compared to the control group as both test substances have contradicting effects in altering these cardiac parameters (*p* > 0.05) (Figure 7A,C–E). Some alteration of cardiac rhythm also observed after co-incubation copper oxide nanoparticles and carbofuran. After co-incubation in copper oxide nanoparticles and carbofuran, the SD1 and SD2 in atrium chamber were significantly higher compared to the single exposure of either copper oxide nanoparticles or carbofuran only (*p* = 0.0014 (SD1) and *p* = 0.0049 (SD2)) (Figure 8A,D), which signifies that copper oxide nanoparticles and carbofuran work synergistically to alter cardiac rhythm and, cause irregular heartbeat in zebrafish larvae. Poincare plot of zebrafish after exposure of copper oxide nanoparticles and carbofuran is shown in Figure A4.

### 2.3. Molecular Docking of Carbofuran with Zebrafish Endogenous AChE

To assess the possible antagonistic action of carbofuran to AChE, molecular docking was performed (Figure 9). AChE is a serine protease responsible for hydrolyzing acetylcholine (ACh) present in the neuromuscular junction and cholinergic brain synapses into choline and acetate [48] This action antagonizes post-synaptic signaling by depleting the ACh pool present in the synaptic cleft, thereby reducing ACh receptor-mediated depolarization of the post-synaptic cell and subsequent nerve impulses.

Molecular docking experiments on ACh and AChE indicated a binding energy of −4.6 kcal/mol. Hydrogen bonding of Ser225 and Gly143 to the carbonyl and the ester oxygen was observed, respectively. Glu224, Trp108, and Tyr355 was observed to have formed attractive charges and π-cation bonds with the quaternary ammonium of the choline group. Trp108 also participated in π-sigma bonds with two of the choline methyl groups. Lastly, His495 showed a hydrophobic interaction with the other choline methyl group (Figure 9A,B).

Molecular docking experiments on carbofuran and AChE showcased high binding affinity of the ligand with a binding energy of −8.0 kcal/mol. Interestingly, carbofuran was observed to bind at a different site than ACh and therefore presented with different interacting residues. Hydrogen bonding of Ser316 and Arg314 to the carbonyl and amide groups was observed, respectively. Trp304, Tyr146, and Phe94 interacted with the benzene ring through π–π stacked interactions. Trp304, Phe94, and Phe315 was also observed to form π–alkyl interactions, while Arg314 was observed to form an additional hydrophobic interaction (Figure 9C,D).

## 3. Discussion

Based on the LC_50_ data, we selected sub-lethal concentration for the cardiac physiology testing. In the case of copper oxide nanoparticles, we selected a concentration around the value of LC_50_, however for carbofuran exposure, the test concentration selected was lower compared to the LC_50_ value of carbofuran. While among the advantages of this method is the use of a simple instrument to check the heart chamber, this technique requires the fish to be in a straight position at the top of the petri dishes to make accurate measurements while incubation in high concentrations (>4.5 µM) of carbofuran may cause severe morphological changes (data not shown). These conditions make it harder to obtain accurate measurements of cardiac physiology and the data accountability would be compromised if the experiment continued, and so the carbofuran concentration was lowered to the point where accurate measurements are still achievable. Several malformation cases after exposure to carbofuran have been reported in several animals like *Hynobius leechii* (Korean salamander), *Bufo gargarizans* (Asiatic toad), and even in zebrafish, which is in line with the findings of this study [16,49,50].

In this study, we found the waterborne exposure to copper oxide nanoparticles could impair contractibility of heart muscles to pump blood. A similar effect was also observed in Sprague–Dawley rats that were given copper nanoparticle synthetized from *Cistus incanus* [48]. Although some studies suggest that copper nanoparticles at low doses work as a cardioprotectant against myocardial infarction [51], high amounts of copper can cause oxidative stress and injure heart muscles [48,52]. Copper nanoparticles works by inducing lysosomal dysfunction which causes irreversible oxidation of proteins, lipids, and nucleic acids, consequently triggering DNA damage and cell death leading to impairment of heart muscle contractility [53,54]. Furthermore, a study also found that copper nanoparticle exposures inhibit AChE activity, which can result in bradycardia although this effect is not specific to cardiac muscles [55].

The toxicity of copper nanoparticles stems from its ability to induce generation of ROS with hydroxyl radicals being the main ROS generated [56]. The toxicity does not only come from the leached copper ion from the nanoparticle, but also from the copper nanoparticle itself. Martinez and co-workers showed that copper nanoparticle which separated from the leached copper ion was highly toxic to DNA. This damage was due to the contact of DNA with the surface area of the copper nanoparticle itself via Fenton-like and Haber–Weiss reactions and the order of magnitude was especially higher in the presence of ascorbate ions [56].

Acetylcholine (ACh) is one of neurotransmitter that is part of autonomous nervous system and mainly located in brain and neuromuscular junction. In brain, ACh perform an important role in the enhancement of alertness, attention, and in learning and memory [57]. In neuromuscular junction, ACh will bind to nicotinic ion channel receptor located in muscle cell membrane and open the ion channel allowing the sodium ions to enter the muscle cells and initiating the sequence of muscle contraction. Acetylcholinesterase (AChE) is an enzyme that produced to breakdown the acetylcholine to prevent the activation of nearby receptor thus terminating the sequence [58]. Previous studies suggest that carbofuran works through inhibition of AChE by disturbing the active site of the enzyme. The inhibition of AChE causes an increase in acetylcholine, which if continuously persistent, will cause hyperexcitation resulting to metabolic stress in tissue. This stress can cause generation of oxidative stress due to the generation of ROS. Overly generated ROS can cause severe damage to the tissue itself especially in sensitive tissues [59]. In this study, alteration of cardiac rhythm in zebrafish larvae was noted at high concentrations of carbofuran exposure. Cardiac rhythm is controlled by the central nervous system by regulating the sympathetic and parasympathetic nervous systems located in the brain and pacemaker [60,61,62]. Based on these assumptions, the arrhythmia observed after carbofuran exposure might have been due to the damage in the central nervous system or pacemaker caused by ROS accumulation in the tissues [63]. Furthermore, like humans, the existence of pacemaker cells was also confirmed in zebrafish, wherein damage in these cells could impact the heart’s ability to maintain sinus rhythm [64]. Thus, we hypothesize that the cardiac damage come from the elevation of oxidative stress cause by copper oxide nanoparticle and the reduction of AChE activity caused by carbofuran (Figure 10).

Molecular docking studies elucidated the binding mechanism of carbofuran as compared to ACh. The AChE active site is subdivided into the catalytic machinery and a choline-binding pocket, termed as the esteratic site and an anionic site, respectively [64]. The esteratic site contains a serine residue directly implicated in the hydrolysis of ACh, which in our case could be Ser225. However, binding of carbofuran did not occur directly within the same binding site as ACh, which therefore presented with different interacting residues. Thus, carbofuran may act as a non-competitive inhibitor in zebrafish AChE which binds to an allosteric site where it may induce conformational changes that render the ACh binding site inactive [64]. Inhibition of AChE would therefore cause an increase in ACh levels in the synaptic cleft. In cardiac muscles, ACh binds to muscarinic ACh receptors which opens ACh-activated K^+^ channels that further polarizes the membrane [64]. This polarization slows down the firing of action potentials and is thus implicated in a slower hear rate.

The toxicity effect of pollution usually comes from exposure to multiple substances. Within the limit of the author’s knowledge, this is the first work that shows the synergistic effect of copper oxide nanoparticles and carbofuran, which causes a disruption in the cardiovascular system in zebrafish. The synergistic effect of heavy metals and pesticides have been explored in several studies. For example, combined exposure to Cd and ethanol increased the level of norepinephrine in the hypothalamus of rats [65]. A similar study also revealed that a combination of Cd and propoxur altered immune and neurotoxicological functions in Wistar rats [66]. Furthermore, a recent study also suggested that a combination of nickel and buprofexin led to very robust damage in zebrafish embryo by increasing oxidative stress, which suggest that a combination of heavy metals and pesticides could become a major threat to the aquatic environment [67]. Other studies of the synergistic effect of copper and pesticides on aquatic animal are summarized in Table A1.

Although carbofuran was known as a highest toxic carbamate, it was easily degrade especially by microbial degradation into carbofuran phenol by hydrolysis [68]. Up until this study was done, no detailed study has done previously to unveil the molecular interaction between copper nanoparticle and carbofuran. However, previous study by Mustapha et al. show that copper can inhibit the degradation of carbofuran, which related to the effect of copper in inhibiting hydrolysis enzyme [69,70,71]. This effect might make the carbofuran present longer in the body and potentially cause more damage.

## 4. Material and Methods

### 4.1. Animal Maintenance and Chemical Exposure

AB strain zebrafish, which used as vertebrate models, was obtained from Taiwan Zebrafish Stock Center at Academia Sinica (TZCAS) and maintained in a continuous aerated water system. The temperature was maintained at 26 °C with 10/14 h of dark/light cycle. Two males and one female adult zebrafish was put into the breeding chamber at night during collection of eggs. In the following morning, the separator was opened and the eggs were collected after two hours. The eggs were then incubated at 28 °C until 24 h post fertilization (hpf). At 24 hpf, the egg was continuously exposed to the test substance/s and at three days post fertilization (dpf) while the cardiac parameters were observed. All protocols and procedures involving zebrafish were approved by the Committee for Animal Experimentation of the Chung Yuan Christian University (Approval No. 109001, issue date 15 January 2020).

### 4.2. Chemical

Copper oxide nanoparticles (20% *w*/*v*) were purchased from Hangzhou Zhiti Purification Technology Co., Ltd. (Hangzhou, China). Stock solutions were diluted using double distilled water into 10mM working solution and sonicated for 20 min before use. Carbofuran was purchased from ANPEL Laboratories Technologies (Shanghai, China) and diluted using double distilled water.

### 4.3. Characterization of Copper Nanoparticle Properties

XRD was carried out to reveal the crystalline phase of the aqueous copper oxide suspension sample. A few drops of the suspension were vacuum-dried on a silicon substrate to carry out XRD analysis. Bruker D8 Advanced eco (Billerica, MA, USA) with a CuK-α radiation source was used to acquire the XRD pattern. The scanning condition used an angular range of 20–80° (2θ), a 0.02° step size, and 5 s per step. Raman spectra were measured using a microscopic Raman system (RAMaker, Protrustech Co., Ltd., Tainan, Taiwan) with excitation wavelengths at 532 nm. The scanned spectral range was 200 to 800 cm^−1^ with a spectral resolution of 1 cm^−1^ and accumulation times were typically every 50 s with a total of 5 scans per spectrum. As reference for calibration (520 cm^−1^), a silicon standard sample was measured. The test sample was prepared by dripping a small quantity of the aqueous solution onto a silicon substrate. Raman spectrum of air-dried sample was collected at random points under 20× objective. FTIR spectroscopy in the range 4000 to 400 cm^−1^ was recorded using a Jasco FTIR-6700 spectrometer (Jasco Co., Tokyo, Japan) on a KBr disc. The Jasco V-770 double beam spectrophotometer (Jasco Co., Tokyo, Japan) was also used for recording UV-Vis spectrum of the test sample to detect absorption. The diluted solution sample was scanned for wavelength range from 200 to 900 nm. The shape and size of the test sample were examined using TEM images. TEM analysis was performed using a Jeol JEM-3010 TEM (JEOL, Tokyo, Japan), operating in TEM mode at 300 kV. TEM sample was prepared by dropping approximately 0.1 mL of diluted dispersions on Ni grids with a carbon film. The size distribution of the sample was obtained by measuring 100 particles, using ImageJ software. The SZ-100 DLS system (Horiba Ltd., Kyoto, Japan) was employed to measure particle size and zeta potential at 25 °C after the suspension was homogenized using high-intensity ultrasonic waves for a period of 15 min. Triplicate measurements were performed for the test sample. DLS is suited to nanoparticles with a sharp size distribution (polydispersity index, PdI < 0.5). To obtain the hydrodynamic diameter (D_h_) of the copper oxide particles, the hydrodynamic radius (R_h_) was calculated by using the Stokes–Einstein equation [72].

### 4.4. Acute Toxicity Test

Acute toxicity tests in zebrafish larvae were done according to OECD Guidelines Section 2 No. 203. Ten zebrafish eggs aged 24 hpf were placed into 3.5 cm petri dish and the desired test concentrations of carbofuran (0.00452, 0.0452, 0.452, 4.52, and 45.2 µM) and copper nanoparticles (0.2, 0.4, 0.6, 0.8, 1, 1.2, 1.4, 1.6, 1.8, and 2 mM) were administered. After exposure, petri dishes were transferred into the incubator at 26 °C under 10/14-h dark/light regime. Mortality rate was documented every 24 h at 48, 72, 96, and 120 hpf and dead fishes were removed at every examination. To check the effect of carbofuran-copper oxide nanoparticle combinations to the survival rate of the larvae, several concentrations were selected and checked based on the result of the acute toxicity test by following the same protocol. Th schematic diagram showing the cardiac performance experimental design is illustrated in Figure A1.

### 4.5. Cardiac Performance Assessment

Zebrafish larvae at 72 hpf were set in 3% methylcellulose to keep the fishes stable during recording. The heart chamber was recorded with high-speed digital charged coupling device (CCD; AZ Instrument, Taichung City, Taiwan) camera mounted on inverted microscope (ICX41, Sunny Optical Technology, Zhejiang, China) capable of recoding at 200 frames per second (fps) for 10 s. Hoffman modulation contrast objective lens with 40× magnification was used to record the zebrafish heart chamber with high quality which focused on the ventricle position, while Lplan objective lens with 10× magnification was used to record the cardiac rhythm. Stroke volume, cardiac output, ejection fraction, and shortening fraction were calculated to assess the cardiac performance. Furthermore, we also assessed the cardiac rhythm parameter, which consisted of the heart rate, atrium-ventricle relaxation interval, ventricle–atrium relaxation interval by getting the timing of diastolic cycle of zebrafish heart using Time Series Analyzer plug-in on ImageJ Software (https://imagej.nih.gov/ij/plugins/time-series.html, accessed on 4 July 2021) and the heart beat variability by calculating the sd1 and sd2 of both heart chamber generated from a Poincare plot using OriginLab Software (Originlab Corporation, Northampton, MA, USA). About 7 fishes were used for each treatment and experiments were performed in triplicates. All analyses were adapted from previously reported methods [28].

### 4.6. Molecular Docking of Carbofuran with Zebrafish Endogenous AChE

#### 4.6.1. Protein Preparation

The three-dimensional structure for zebrafish (*Danio rerio*) AChE is unavailable in the RCSB Protein Data Bank, therefore structural homology was done to obtain the enzyme structure. An amino acid sequence of *D. rerio* AChE was retrieved from the Universal Protein Resource Database (Accession Number Q9DDE3) and homology modeling was done using the SWISS-MODEL workspace. Template 1q83.1.A was selected for protein modeling with sequence identity of 61.55%. Model quality was given by Qmean (score of −1.55) and structural assessment was done via Ramachandran plot (Figure A2). The 3D structure was added as PDB format to the UCSF Chimera platform [37,38]. Non-standard residues were then removed. Chain A was left from the dimer, removing Chain B. The enzyme was subsequently minimized by the conjugate gradient method (10 steps-steps size 0.02 Å) and the steepest descent method (100 steps-step size 0.02 Å).

#### 4.6.2. Ligand Preparation

Acetylcholine and carbofuran were used as ligands. Their respective structures were drawn using ChemDraw Professional 16.0 and converted to SMILES format. The SMILES file of each ligand was then converted to SYBYL mol2 using Avogadro software version 1.2.0. The SYBYL mol2 files were then imported into UCSF Chimera for docking [39].

#### 4.6.3. Molecular Docking

The minimized pdb file of AChE and the mol2 files of ACh and carbofuran were docked using UCSF Chimera version 1.15 [37]. Missing hydrogen atoms and charges were added using the Gasteiger charge method through Amber’s Antechamber module [40,41]. The protocol for flexible ligands into flexible active site was utilized in the docking procedure to allow the ligand’s rotational and translational maneuver within the specified grid 8. Autodock Vina’s Broyden–Fletcher–Goldfarb–Shanoo (BFGS) algorithm version 1.5.6 was utilized for the virtual screening of the library [42]. Lastly, binding scores were determined using UCSF chimera, wherein the best conformation was visualized in Biovia Discovery Studios version 21.1.0.

### 4.7. Statistical Analysis

Statistical data were generated using GraphPad Prism software package (GraphPad Software version 8 Inc., La Jolla, CA, USA). To determine the level of significance, parametric and non-parametric test was performed according to the normality and data distribution. The degree of significance was set at *p-*value < 0.05.

## 5. Conclusions

This study explored the potential cardiotoxicity after copper oxide nanoparticles and carbofuran exposure in zebrafish embryos for the first time. Our work supports the idea that copper oxide nanoparticles and carbofuran work synergistically to increase toxicity in the cardiac system of zebrafish showing irregular heartbeats after waterborne toxicant exposure. In addition, molecular docking experiments showed that carbofuran can inhibit in silico the activity of AChE thus altering cardiac performance observed in zebrafish larvae.

## Figures and Tables

**Figure 1 ijms-22-08259-f001:**
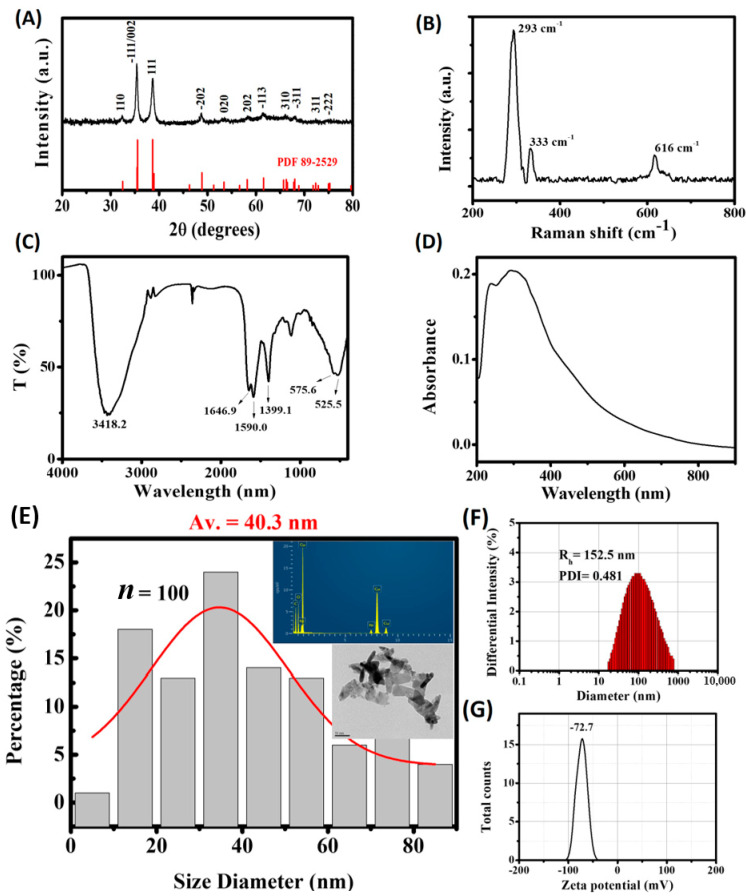
Analysis of physical properties of copper oxide nanoparticles. (**A**) X-ray diffraction (XRD) analysis; (**B**) Raman spectrum; (**C**) Fourier-transform infrared (FTIR) spectrum; (**D**) UV-Vis absorption spectrum; (**E**) Transmission electron Microscopy (TEM) observation and energy-dispersive X-ray spectroscopy (EDS); (**F**) Dynamic light scattering (DLS) analysis; (**G**) Zeta potential measurement.

**Figure 2 ijms-22-08259-f002:**
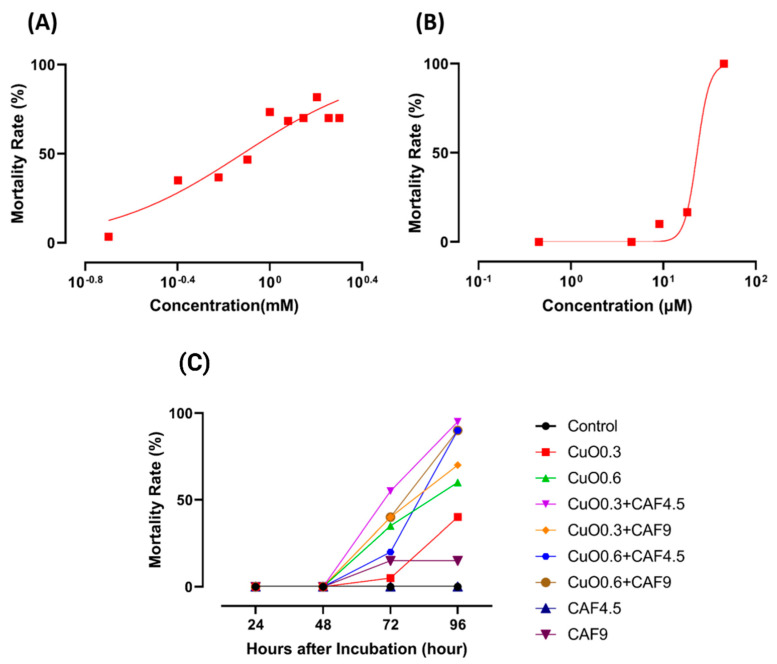
Mortality rate of zebrafish embryos after incubation in several concentration of either (**A**) CuO nanoparticles, (**B**) carbofuran or (**C**) combination of both compounds (CuO = mM, CAF = µM).

**Figure 3 ijms-22-08259-f003:**
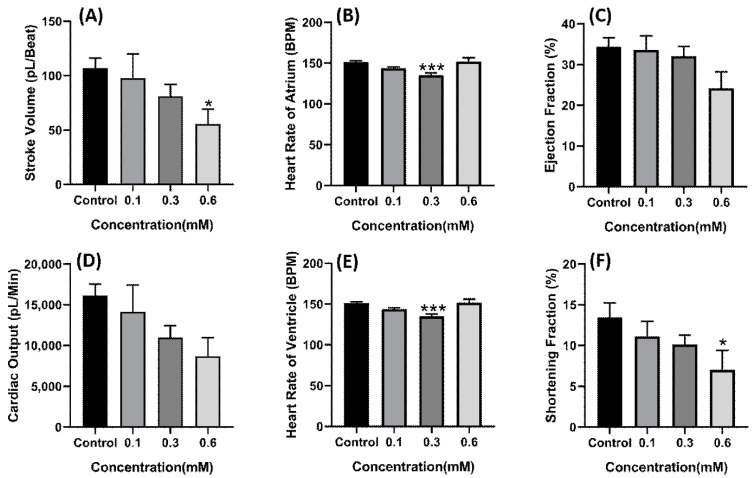
Cardiac physiology alteration induced by CuO nanoparticle exposure in zebrafish larvae. (**A**) Stroke volume. (**B**) Heart rate of atrium. (**C**) Ejection fraction. (**D**) Cardiac output. (**E**) Heart rate of ventricle. (**F**) Shortening fraction. Data were statistically analyzed using ordinary one-way ANOVA with the Dunnett post-hoc test for multiple comparisons. * *p* < 0.05, *** *p* < 0.001.

**Figure 4 ijms-22-08259-f004:**
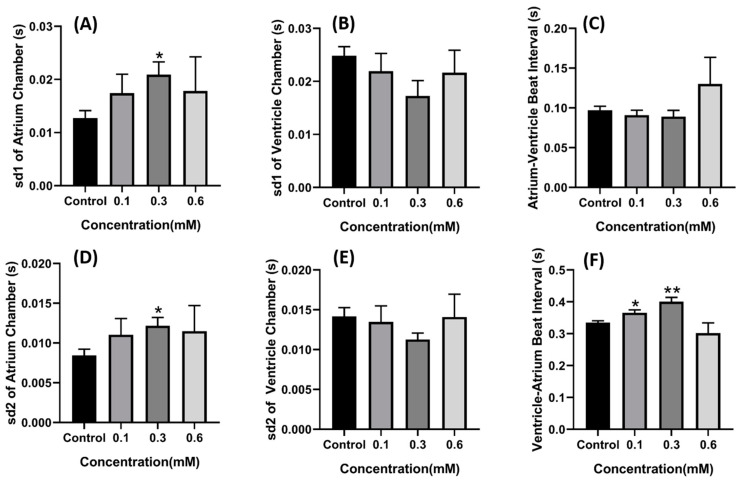
Cardiac rhythm alteration induced by CuO nanoparticle exposure in zebrafish larvae. (**A**) Atrium SD1. (**B**) Ventricle SD1. (**C**) Atrium–ventricle interval. (**D**) Atrium SD2. (**E**) Ventricle SD2. (**F**) Ventricle–atrium interval. Data were statistically analyzed using a Brown–Fosythe one-way ANOVA with Dunnett post-hoc test for multiple comparisons. * *p* < 0.05, ** *p* < 0.005.

**Figure 5 ijms-22-08259-f005:**
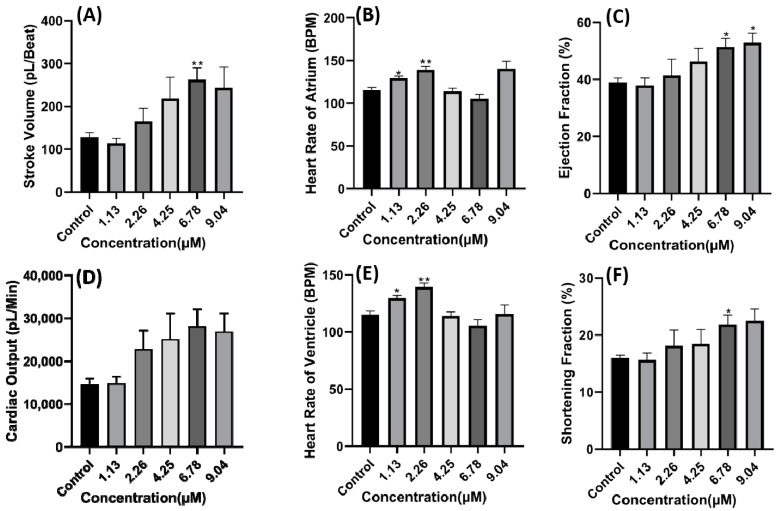
Cardiac physiology alteration induced by carbofuran in zebrafish larvae. (**A**) Stroke volume. (**B**) Heart rate of atrium. (**C**) Ejection fraction. (**D**) Cardiac output. (**E**) Heart rate of ventricle. (**F**) Shortening fraction. Data were statistically analyzed using ordinary one-way ANOVA with a Dunnett post-hoc test for multiple comparisons. * *p* < 0.05, *** p* < 0.005.

**Figure 6 ijms-22-08259-f006:**
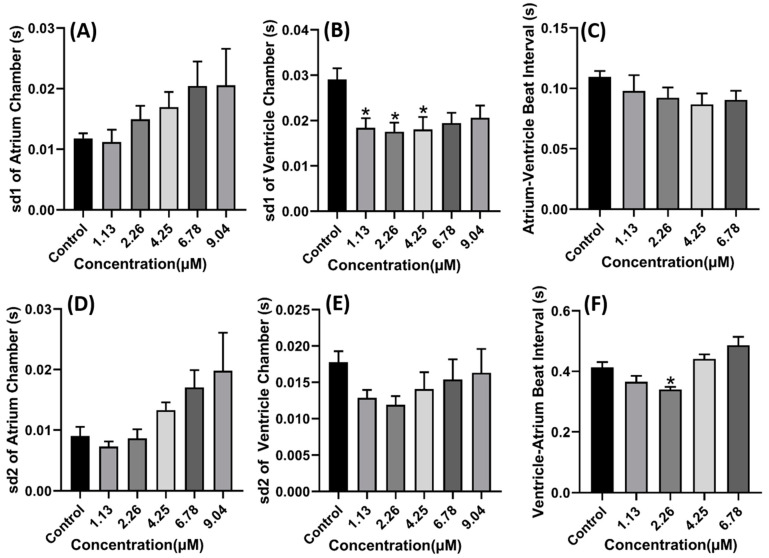
Cardiac rhythm alteration induced by carbofuran in zebrafish larvae. (**A**) Atrium SD1. (**B**) Ventricle SD1. (**C**) Atrium–ventricle interval. (**D**) Atrium SD2. (**E**) Ventricle SD2. (**F**) Ventricle–atrium interval. Data were statistically analyzed using a Brown–Fosythe one-way ANOVA with a Dunnett post-hoc test for multiple comparisons. * *p* < 0.05.

**Figure 7 ijms-22-08259-f007:**
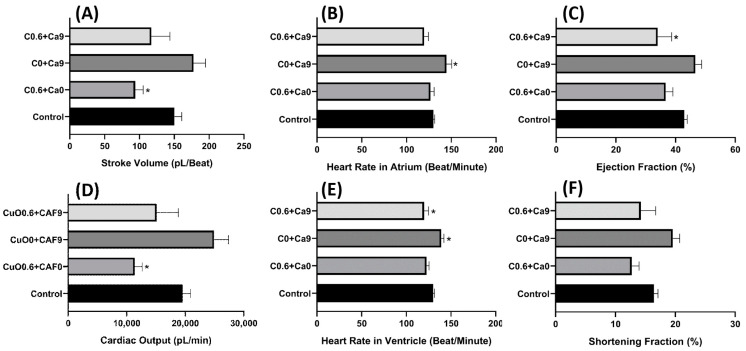
Cardiac physiology alteration induced by CuO nanoparticle and carbofuran co-incubation in zebrafish larvae. (**A**) Stroke volume. (**B**) Heart rate of atrium. (**C**) Ejection fraction. (**D**) Cardiac output. (**E**) Heart rate of ventricle. (**F**) Shortening fraction. Data were statistically analyzed using the ordinary one-way ANOVA test with the Dunnett post-hoc test for multiple comparison analysis. * *p* < 0.05. (CuO = copper nanoparticle (mM), CAF = carbofuran (µM)).

**Figure 8 ijms-22-08259-f008:**
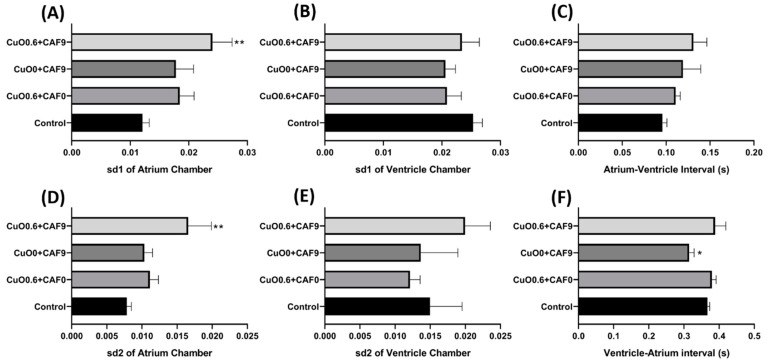
Cardiac rhythm alteration induced by CuO nanoparticle and carbofuran co-incubation in zebrafish larvae. (**A**) Atrium SD1. (**B**) Ventricle SD1. (**C**) Atrium–ventricle interval. (**D**) Atrium SD2. (**E**) Ventricle SD2. (**F**) Ventricle–atrium interval. Data were statistically analyzed using the Kruskal–Wallis ANOVA test with Dunn’s post-hoc test for multiple comparison * *p* < 0.05, *** p* < 0.005. (CuO = copper nanoparticle (mM), CAF = carbofuran (µM)).

**Figure 9 ijms-22-08259-f009:**
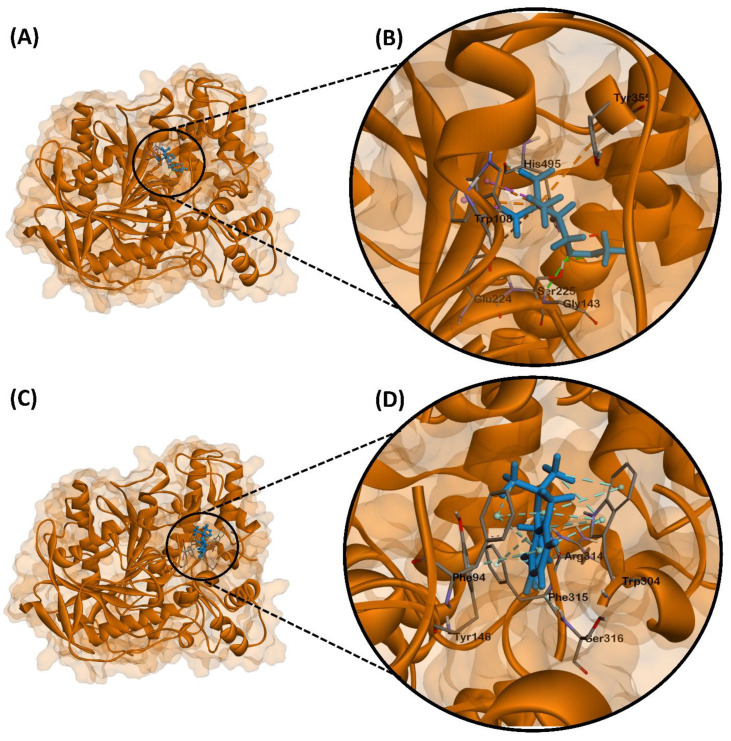
Molecular docking poses comparing the binding mechanism of ACh and carbofuran to AChE chain A. (**A**) Full view of the ACh-AChE complex, (**B**) expanded view of ACh’s binding site with interacting residues, (**C**) full view of the carbofuran-AChE complex, (**D**) expanded view of carbofuran’s binding site with interacting residues.

**Figure 10 ijms-22-08259-f010:**
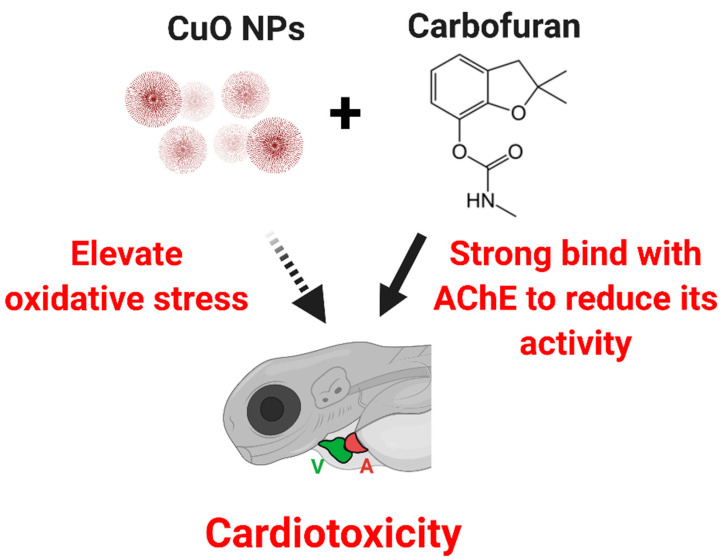
Proposed model for cardiotoxicity triggered by CuO nanoparticle and carbofuran co-incubation in zebrafish larvae (A = atrium, V = ventricle). The solid line indicates direct evidence provide by our molecular docking data, while dotted line indicates indirect evidence collected from literature.

## Data Availability

Original data and videos can be obtained from authors upon request.

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
