# Peer review of "Co-Treatment of Copper Oxide Nanoparticle and Carbofuran Enhances Cardiotoxicity in Zebrafish Embryos"

_ijms, 2021, doi:10.3390/ijms22158259_

Round 1

Reviewer 1 Report

Title: Co-treatment of Copper Oxide Nanoparticle and Carbofuran Enhances Cardiotoxicity in Zebrafish Embryos

Authors: Saptura, F.; Chen, K.; Lee, J.; Hung, S.; Huang, J.; Pang, Y.; Munoz, J.; Macabeo, A.; Uapipatanakul, B.; Hsiao, C.

Overview:

The manuscript studies the effects of  CuO nanoparticles and carbofuran exposure on the embryos of zebrafish; their combined effects were also studied. Mortality was initially assessed, followed by assessment of cardiac output at sub-lethal doses.. Nanoparticle size, shape, and structure are characterized. Molecular docking experiments between carbofuran and AChe were also conducted. Mechanisms. Overall, the study subject of waterborne pollutants is an area of concern. The observation of an interaction between the nanoparticles and carbofuran is  interesting. Though the work is generally well written, there are minor language errors present throughout.

Minor Concerns

Introduction:

  1. Line 41: “ultimately comes back to humans in food forms” is awkwardly worded.
  2. Minor language errors. E.g., Line 42: “... pesticides can cause adverse effects to aquatic animals.” should be changed to “... pesticides can have adverse effects on aquatic animals”.

Results:

  1. Line 87: possible character errors “2@@=”.
  2. Line 92: “Fishing out” should be changed to 'isolating'.
  3. Minor language errors. E.g., Line 100 “... The absorption […] is a characteristic stretching vibrations ...“ should be changed to “... the absorption […] is a characteristic of stretching vibrations ...”

Discussion:

  1. Minor language errors. E.g.., Line 267: “we found the waterborne exposure of ...” should be changed to “we found that waterborne exposure to ...”
  2. Ach/AChE should be defined.

Materials and methods:

  1. Minor language errors. E.g., Line 333:  “AB strain zebrafish was used as vertebrate model and maintained ...” should be changed to “AB strain zebrafish were used as vertebrate models, and they were maintained ...”

Author Response

Minor Concerns

Introduction:

  1. Line 41: “ultimately comes back to humans in food forms” is awkwardly worded.

Thank you for the correction. The sentence in line 40 and 41 has been changed to make it more natural in the updated manuscript.

  1. Minor language errors. E.g., Line 42: “... pesticides can cause adverse effects to aquatic animals.” should be changed to “... pesticides can have adverse effects on aquatic animals”.

Thank you for the correction. The sentence in line 41 and 42 has been changed according to reviewer suggestion.

Results:

  1. Line 87: possible character errors “2@@=”.

Thank you for the correction. The mistyping in line 87 has been corrected in the updated manuscript.

  1. Line 92: “Fishing out” should be changed to 'isolating'.

Thank you for your suggestion. In order to make the meaning of the sentence clearer, it has been revised in the updated manuscript.

  1. Minor language errors. E.g., Line 100 “... The absorption […] is a characteristic stretching vibrations ...“ should be changed to “... the absorption […] is a characteristic of stretching vibrations ...”

Thank you for the correction. The sentence in line 100 has been changed according to reviewer suggestion.

Discussion:

  1. Minor language errors. E.g.., Line 267: “we found the waterborne exposure of ...” should be changed to “we found that waterborne exposure to ...”

Thank you for the correction. The updated manuscript has been changed according to the reviewe suggestion.

  1. Ach/AChE should be defined.

Thank you for the wonderful suggestion. The authors also agree that some introduction about ACh and AChE will improve the reader comprehension, thus in the updated manuscript. The authors already add some brief introduction about the ACh and AChE.

Materials and methods:

  1. Minor language errors. E.g., Line 333:  “AB strain zebrafish was used as vertebrate model and maintained ...” should be changed to “AB strain zebrafish were used as vertebrate models, and they were maintained ...”

Thank you for the correction. The updated manuscript has been changed according to the reviewer suggestion.

Reviewer 2 Report

In the present manuscript, Saputra and co-workers report that the co-treatment of Copper Oxide Nanoparticle and Carbofuran
Enhances Cardiotoxicity in Zebrafish Embryos. The manuscript is written well but contains some redundant use of unscientific terms and the research design, especially the characterization of NPs has to be improved. I would recommend major revisions based on the following reasons:

  1. The introduction does not provide enough background as to why the study was performed using CuO nanoparticles? Is it the largest known metal nanoparticle pollutant? What about other heavy metal elements?
  2. The nanoparticles have too high polydispersity and it seems that there is a lot of agglomeration in the nanoparticles. Agglomerated nanoparticles could influence their toxicity and biological fate in the species of interest and henceforth well-dispersed nanoparticles must be prepared for this study to compare if the nanoparticle agglomeration determines the toxicity observed in later studies. 
  3. The TEM image clearly indicates that the nanoparticles are an agglomerated mass. Please include images of well-dispersed nanoparticles. 
  4. How do the authors confirm that the TEM image shown is of a Copper nanoparticle? Have they carried out any EDS/EDX on these to identify Copper? From the TEM image, it seems that the nanoparticles are covered with some kind of polymer. There are various publications about synthesis of copper oxide nanoparticles that the authors should refer to for their work (Example: https://doi.org/10.1021/acsbiomaterials.8b01092) and prepare well-dispersed Cu nanoparticles for their study.
  5. It seems that the Cu nanoparticles were purchased from a commercial supplier. It seems like a very poor choice as the nanoparticles are not very good in quality as the size distribution is extremely poor and might result in the observed toxicity effects reported in the manuscript. Please use a modified formulation.

Author Response

Comments and Suggestions for Authors

In the present manuscript, Saputra and co-workers report that the co-treatment of Copper Oxide Nanoparticle and Carbofuran Enhances Cardiotoxicity in Zebrafish Embryos. The manuscript is written well but contains some redundant use of unscientific terms and the research design, especially the characterization of NPs has to be improved. I would recommend major revisions based on the following reasons:

  1. The introduction does not provide enough background as to why the study was performed using CuO nanoparticles? Is it the largest known metal nanoparticle pollutant? What about other heavy metal elements?

Thank you for the comment. The selection of copper nanoparticle was because copper nanoparticle was commonly used in agricultural field due to its availability and antimicrobial activity. The authors agree that other heavy metal nanoparticle that information commonly used in agriculture could increase the reader comprehension, thus some information about commonly used heavy metal nanoparticle in agriculture industries has been added in the revised manuscript (line 62-65).

  1. The nanoparticles have too high polydispersity and it seems that there is a lot of agglomeration in the nanoparticles. Agglomerated nanoparticles could influence their toxicity and biological fate in the species of interest and henceforth well-dispersed nanoparticles must be prepared for this study to compare if the nanoparticle agglomeration determines the toxicity observed in later studies.

Thank you for your valuable suggestion. According the DLS and TEM analyses, the test CuO nanoparticles exhibit a high polydispersity and a lot of agglomeration. As described in some literature, agglomeration/aggregation usually occurred in soil and lead to modifications in the solubility of CuO nanoparticle in soil. When these CuO particles in the soil are discharged, it will be close to true morphology of these particles in the water body. We appreciate your comments, and a future research work on the toxicity may also be performed using the CuO particles with a narrow particle size distribution for comparison.

Dimkpa, C.O., Can nanotechnology deliver the promised benefits without negatively impacting soil microbial life? Journal of Basic Micorbiology 2014, 54, 889-904.

Cornelis, G.; Hund-Rinke, K.; Kuhlbusch, T.; Van den Brink, N.; Nickel, C., Fate and bioavailability of engineered nanoparticles in soils: A review. Crit. Rev. Environ. Sci. Technol. 2014, 44, 2720-2764.

  1. The TEM image clearly indicates that the nanoparticles are an agglomerated mass. Please include images of well-dispersed nanoparticles. 

Thank you for your suggestion. In order to make the particle morphology in the TEM images more representative, the photo in Figure 1E has been replaced in the updated manuscript.

  1. How do the authors confirm that the TEM image shown is of a Copper nanoparticle? Have they carried out any EDS/EDX on these to identify Copper? From the TEM image, it seems that the nanoparticles are covered with some kind of polymer. There are various publications about synthesis of copper oxide nanoparticles that the authors should refer to for their work (Example: https://doi.org/10.1021/acsbiomaterials.8b01092) and prepare well-dispersed Cu nanoparticles for their study.

Thank you for your suggestion. In addition to CuO phase, some weaker peaks of NaNO3 have been detected by XRD in the test CuO nanoparticle samples. As quite similar to your observation, these CuO nanoparticles may be covered by a thin layer of NaNO3 precipitation. With reference to various publications, it is reasonable to speculate that the CuO sample was prepared by electrochemical synthesis technique. In order to confirm the chemical composition of CuO nanoparticles, EDS analysis has been performed after the CuO sample was leached. So, the EDS pattern has also been added into the Figure 1 to show the identity of particles, as seen in the updated manuscript. Thank you again on providing this constructive comment.   

  1. It seems that the Cu nanoparticles were purchased from a commercial supplier. It seems like a very poor choice as the nanoparticles are not very good in quality as the size distribution is extremely poor and might result in the observed toxicity effects reported in the manuscript. Please use a modified formulation.

Thank you for the valuable suggestion. However, the authors not total agree with the reviewer argument as the authors thinks that the Cu nanoparticle was purchase from a reputable company that has an ISO9001 standard and the product has been used in many studies that use nanoparticle as it main subject (Feng et al., 2020; Feng et al., 2019; Shi, Qi, & Zeng, 2020). Although the authors also agree that from the characterization result is seems that the particle distribution was not perfect, it might be come from the rough surface of the nanoparticle itself that is uneven which make it easier for the nanoparticle to agglomerate. (Fabre, Salameh, Ciacchi, Kreutzer, & van Ommen, 2016).

Feng, H.; Lv, L.; Pang, Y.; Wang, Z.; Gao, D.; Zhang, Z. Experimental study on the effects of the fiber and nano-Fe2O3 on the properties of the magnesium potassium phosphate cement composites. journal of materials research and technology 2020, 9, 14307-14320.

Feng, H.; Zhao, X.; Chen, G.; Miao, C.; Zhao, X.; Gao, D.; Sun, G. The effect of nano-particles and water glass on the water stability of magnesium phosphate cement based mortar. Materials 2019, 12, 3755.

Shi, Z.; Qi, S.; Zeng, L. Effect of Nanostructured TiO2 on Rheological Properties of Fresh Cement Slurry. Journal of nanoscience and nanotechnology 2020, 20, 4907-4913.

Fabre, A.; Salameh, S.; Ciacchi, L.C.; Kreutzer, M.T.; van Ommen, J.R. Contact mechanics of highly porous oxide nanoparticle agglomerates. Journal of Nanoparticle Research 2016, 18, 1-13.

Reviewer 3 Report

This study aimed to evaluate the effects of carbofuran and copper oxide nanoparticles on the cardiovascular system of zebrafish and unveil the mechanism behind them.

The study is interesting and provides information about how copper oxide nanoparticle and carbofuran combinations work synergistically to enhance toxicity on the cardiovascular performance of zebrafish larvae. Nevertheless, some issues must be addressed:

The abbreviations should be defined at their first appearance in the text (example “TEM” lines 102, "DLS" line 111, "AchE" line 223).

“The increased need for pesticides in agricultural industries increases has become an increasing concern to the aquatic environment. ” (lines 34-35). It must be rephrased.

In material and methods, the authors could give more information about how they observed the alteration of the cardiac rhythm and the irregular heartbeat in the atrium chamber and in the ventricle.

In the Results section must be presented only the results of the present study. No comments of the results (“Based on the previously reported data of copper” or “ In agreement to our expectations”) should be made here, neither references to other studies.  

The Discussion section can be improved with some of the information given in the Results section.

Which are the limitations of the present study?

Author Response

Comments and Suggestions for Authors

This study aimed to evaluate the effects of carbofuran and copper oxide nanoparticles on the cardiovascular system of zebrafish and unveil the mechanism behind them. The study is interesting and provides information about how copper oxide nanoparticle and carbofuran combinations work synergistically to enhance toxicity on the cardiovascular performance of zebrafish larvae. Nevertheless, some issues must be addressed:

The abbreviations should be defined at their first appearance in the text (example “TEM” lines 102, "DLS" line 111, "AchE" line 223).

Thank you for the reminder. The revised manuscript has been updated to accommodate this problem.

“The increased need for pesticides in agricultural industries increases has become an increasing concern to the aquatic environment. ” (lines 34-35). It must be rephrased.

Thank you for the correction. The mistyping in line 34-35 has been corrected in the updated manuscript.

In material and methods, the authors could give more information about how they observed the alteration of the cardiac rhythm and the irregular heartbeat in the atrium chamber and in the ventricle.

Thank you for the suggestion. The alteration and of the cardiac rhythm and irregular heartbeat was calculated based on the change in the heart rate, atrium to ventricle relaxation interval vice versa, and the heart beat variability which the detail formulation already mentioned in our previously published method (Hsiao et al., 2020). The authors agree with the reviewer that addition of information about the cardiac rhythm and heart rate variability could increase the reader comprehension, thus the manuscript already updated according to reviewer suggestion (line 420-426).

Hsiao, C.-D.; Wu, H.-H.; Malhotra, N.; Liu, Y.-C.; Wu, Y.-H.; Lin, Y.-N.; Saputra, F.; Santoso, F.; Chen, K.H.-C. Expression and Purification of Recombinant GHK Tripeptides Are Able to Protect against Acute Cardiotoxicity from Exposure to Waterborne-Copper in Zebrafish. Biomolecules 2020, 10, 1202.

In the Results section must be presented only the results of the present study. No comments of the results (“Based on the previously reported data of copper” or “ In agreement to our expectations”) should be made here, neither references to other studies.  

Thank you for the valuable suggestion. The “previous reported data” mentioned here was refer to the single exposure data. We agree with the reviewer that it can make misinterpretation, thus the updated manuscript has been updated to accommodate the problem. Furthermore, the comment in the result section has been removed according to the reviewer suggestion. 

The Discussion section can be improved with some of the information given in the Results section.

Thank you for the suggestion. The authors also agree that some discussion part can be improved. Thus in the updated manuscript, we already add a brief introduction about ACh and AChE as the authors think that is the missing part in the discussion part.

Which are the limitations of the present study?

Thank you for the comment. As it already stated, the limitation of this study is that the method used for recording cannot record the fish if severe malformation happened to the fish as it will reduce the accountability of the data which make the authors have to reduce the carbofuran concentration until its possible to record the heart chamber. Thus in the future, new method should be developed to enable recording of cardiac chamber even if the fish have high malformation rate.

Reviewer 4 Report

The authors report that synergic presence of copper oxide nanoparticles and carbofuran pesticide could be cardiotoxic for zebrafishes. The conclusions are based on the monitoring of cardiac activity and onset of malformations or edemas. The work is scientifically sound and might be of interest for a broad audience. I recommend it for publication after minor reviews.

Figure 1: the caption is wrong. Figures from 1A to 1G are not described and info regarding the toxicity of copper oxide nanoparticles have nothing to do with the images.

Figure 12: neither the image nor the caption explain what are the sectors marked in red and green with letters A and V. The figure should be self explanatory.

A DLS measurement (both size and zeta potential) of copper oxide nanoparticles incubated with carbofuran could be useful to understand whether carbofuran could induce further aggregation of the particles or if it could stick to particles surface and be transported more efficiently, hence increasing cardiotoxicity.

Author Response

Comments and Suggestions for Authors

The authors report that synergic presence of copper oxide nanoparticles and carbofuran pesticide could be cardiotoxic for zebrafishes. The conclusions are based on the monitoring of cardiac activity and onset of malformations or edemas. The work is scientifically sound and might be of interest for a broad audience. I recommend it for publication after minor reviews.

 Figure 1: the caption is wrong. Figures from 1A to 1G are not described and info regarding the toxicity of copper oxide nanoparticles have nothing to do with the images.

Thank you for the correction. The wrong figure has been corrected in the updated manuscript.

Figure 12: neither the image nor the caption explain what are the sectors marked in red and green with letters A and V. The figure should be self explanatory.

Thank you for the valuable suggestion. The authors also agree that the figure should be self-explanatory, Thus the explanation for A and V in figure 12 has been added in the figure legend.

A DLS measurement (both size and zeta potential) of copper oxide nanoparticles incubated with carbofuran could be useful to understand whether carbofuran could induce further aggregation of the particles or if it could stick to particles surface and be transported more efficiently, hence increasing cardiotoxicity.

Thank you for the valuable suggestion. However, we found it was difficult to conduct this experiment due to COVID-19 lockdown. We appreciate your comments and already did some discussion in this revised paper according to reviewer’s comments (line 345-351).

Round 2

Reviewer 2 Report

The authors do not provide sufficient evidence about the nanoparticle nature of the CuO particles used in the experiment. The TEM image is not clear. Also, what kind of grids were used by the authors? Most TEM grids are made of Copper and hence the EDS will show peaks for Copper. Have the authors used a non-copper grid? Such as nickel or other kind of grids? Also, the resolution of the TEM image is very very poor. Please improve it by using an image of higher resolution. Also, why is the Copper in the EDS spectra labeled as "CU" and not "Cu"? Please modify.

Author Response

Comments and Suggestions for Authors

The authors do not provide sufficient evidence about the nanoparticle nature of the CuO particles used in the experiment. The TEM image is not clear. Also, what kind of grids were used by the authors? Most TEM grids are made of Copper and hence the EDS will show peaks for Copper. Have the authors used a non-copper grid? Such as nickel or other kind of grids? Also, the resolution of the TEM image is very very poor. Please improve it by using an image of higher resolution. Also, why is the Copper in the EDS spectra labeled as "CU" and not "Cu"? Please modify.

Thank you for the valuable suggestion. Cu grid has been replaced by Ni grid for TEM and EDS analysis. Another TEM instrument (Jeol JEM-3010, Tokyo, Japan), with higher resolution was employed to make the CuO anparticle morphology image more representative and clear than previous version. Therefore, the particle size distribution and its average particle size are also revised. The plot and photos in Figure 1E have been replaced in the updated manuscript. Thus, TEM information and some CuO nanoparticle morphology-related content have also been altered, as shown in Line 115, 116, 120, 121, 122, 393, 394, 395, and 396, respectively. We are very grateful for your comments and have made some corrections in this revised paper according to reviewer’s comments.

Reviewer 3 Report

The authors did the suggested improvements to the manuscript.

Author Response

The authors did the suggested improvements to the manuscript.

Thank you again on providing us constructive comments

Round 3

Reviewer 2 Report

Authors have addressed the comments from the reviewer.